# Comparison of Machine Learning Models for Brain Age Prediction Using Six Imaging Modalities on Middle-Aged and Older Adults

**DOI:** 10.3390/s23073622

**Published:** 2023-03-30

**Authors:** Min Xiong, Lan Lin, Yue Jin, Wenjie Kang, Shuicai Wu, Shen Sun

**Affiliations:** 1Department of Biomedical Engineering, Faculty of Environment and Life, Beijing University of Technology, Beijing 100124, China; xiongmin@emails.bjut.edu.cn (M.X.); jinyue@emails.bjut.edu.cn (Y.J.); s201915701@emails.bjut.edu.cn (W.K.); wushuicai@bjut.edu.cn (S.W.); sunshen@bjut.edu.cn (S.S.); 2Intelligent Physiological Measurement and Clinical Translation, Beijing International Base for Scientific and Technological Cooperation, Beijing University of Technology, Beijing 100124, China

**Keywords:** brain age prediction, machine learning, multi-modality MRI, UK Biobank

## Abstract

Machine learning (ML) has transformed neuroimaging research by enabling accurate predictions and feature extraction from large datasets. In this study, we investigate the application of six ML algorithms (Lasso, relevance vector regression, support vector regression, extreme gradient boosting, category boost, and multilayer perceptron) to predict brain age for middle-aged and older adults, which is a crucial area of research in neuroimaging. Despite the plethora of proposed ML models, there is no clear consensus on how to achieve better performance in brain age prediction for this population. Our study stands out by evaluating the impact of both ML algorithms and image modalities on brain age prediction performance using a large cohort of cognitively normal adults aged 44.6 to 82.3 years old (N = 27,842) with six image modalities. We found that the predictive performance of brain age is more reliant on the image modalities used than the ML algorithms employed. Specifically, our study highlights the superior performance of T1-weighted MRI and diffusion-weighted imaging and demonstrates that multi-modality-based brain age prediction significantly enhances performance compared to unimodality. Moreover, we identified Lasso as the most accurate ML algorithm for predicting brain age, achieving the lowest mean absolute error in both single-modality and multi-modality predictions. Additionally, Lasso also ranked highest in a comprehensive evaluation of the relationship between BrainAGE and the five frequently mentioned BrainAGE-related factors. Notably, our study also shows that ensemble learning outperforms Lasso when computational efficiency is not a concern. Overall, our study provides valuable insights into the development of accurate and reliable brain age prediction models for middle-aged and older adults, with significant implications for clinical practice and neuroimaging research. Our findings highlight the importance of image modality selection and emphasize Lasso as a promising ML algorithm for brain age prediction.

## 1. Introduction

As the world’s population ages and the prevalence of dementia rates rises, early detection, prevention, and treatment of neurological aspects of aging, such as cognitive decline and dementia, are becoming increasingly important. The degree of deviation from the normal range is an indication of pathological brain aging. This has fueled a growing interest in the development of methods to identify individuals deviating from a normative brain aging trajectory. The concept of brain age, an estimated biological age from anatomical and/or functional brain imaging data, has garnered significant attention in recent years [1,2]. Predictive deviations of brain age from chronological age have led to the development of personalized biomarkers for describing healthy brain development, abnormal aging, and early signs of clinical neuropsychiatric issues [3]. Brain age prediction using machine learning (ML) techniques can infer an individual’s brain age from neuroimaging data, where brain age is roughly equivalent to the underlying biological age of the brain. Once trained, the brain-age model can be used to assess brain health in independent samples. Individuals with an estimated brain age below their chronological age have younger brains than their age-matched, healthy contemporaries, indicating a greater resistance to pathology and neurodegeneration. Conversely, accelerated brain aging occurs when predicted brain age exceeds chronological age, suggesting the brain has been subjected to cumulative insults or severe pathological impacts. The brain age gap estimation (BrainAGE) [4,5] metric has been introduced as an alternative to determine the degree of neuropathology, defined as the difference between the predicted brain age and chronological age. BrainAGE research using neuroimaging data has yielded important insights into the pathology of the brain in a wide range of neurological diseases such as Alzheimer’s disease (AD) [6], mild cognitive impairment [6], traumatic brain injury [7], epilepsy [8], multiple sclerosis [9], as well as psychiatric disorders such as schizophrenia [10], bipolar disorder [11], and major depressive disorder [12].

Various ML algorithms, such as the last absolute shrinkage and selection operator (Lasso) [13,14,15], relevance vector regression (RVR) [1,4,16,17], support vector regression (SVR) [14,18,19,20,21,22], multilayer perceptron (MLP) [23], and extreme gradient boosting (XgBoost) [24,25], have been employed for predicting brain age by using relevant features extracted from neuroimages. One such feature is the gray matter (GM) density map, which has been utilized in several studies to predict brain age. Franke et al. [4] developed a brain age prediction system using a RVR approach based on preprocessed gray matter (GM) density maps. Their system achieved a mean absolute error (MAE) of 4.98 years after training on a cohort of 410 healthy adults aged 20 to 86 years. Similarly, Le et al. [19] applied the SVR method to GM density maps from a larger cohort of 964 individuals aged 18 to 60 years and obtained a similar MAE of 4.84 years. Varikuti et al. [15] took a different approach, using non-negative matrix factorization (NMF) clustering and Lasso regression analysis to develop a brain age prediction model based on GM density maps from 693 older individuals (aged 55 to 75 years), achieving an impressive MAE of 3.6 years. In addition to good performance, the model produced neurobiologically interpretable maps. The combination of deformation fields with GM volume has been shown to improve the accuracy of brain age prediction. For example, a RVR approach trained on a large cohort of healthy individuals (aged 20–86 years) achieved better results (MAE = 6.90 years) than using GM volumetric information alone (MAE = 7.96 years). [16]. Transfer learning is another approach that has been employed to improve brain age prediction accuracy. Lin and colleagues [26] utilized transfer learning to extract features from 594 healthy older individuals aged 50 to 90 years, which were then used as input for RVR. This approach achieved an MAE of 4.51 years. Diffusion tensor imaging (DTI) has been widely used to investigate WM microstructure, providing important insights into brain aging. Mwangi et al. [17] applied the RVR approach to a cohort of 188 participants aged 4 to 85 years using popular DTI metrics and found that fractional anisotropy ageing patterns follow non-linear trajectories. In addition to structural changes, alterations in brain structural and functional connectivity have also been examined for brain age prediction. For example, Lin et al. [23] employed MLP to predict brain age based on the structural connectivity network of 112 healthy participants aged 50.4–79.1 years and reported a mean MAE of 4.29 years. Resting-state functional MRI (rsfMRI) has also been utilized for brain age prediction. Vergun and colleagues [22] used a SVR algorithm trained on rsfMRI data from 117 healthy individuals (aged 19–85 years) and found that SVR with a linear kernel performed better than a Gaussian kernel. Multi-modal imaging, which combines different types of imaging techniques, has been shown to provide complementary information and improve the accuracy of predicting brain age. For instance, Anatürk et al. [24] utilized T1-weighted MRI (T1), DTI, and T2 fluid attenuated inversion recovery (T2) MRI to extract 1118 GM features and 245 WM features from 537 participants aged 60.34 to 82.76 years. They applied the XgBoost model and achieved an impressive MAE of 3.32 years. De Lange et al. [25] further explored the potential of multi-modal imaging for brain age prediction. They employed an XgBoost algorithm trained on T1, DTI, and rsfMRI image modalities from 610 participants aged 60.34 to 84.58 years. Their results demonstrated that combining the three modalities was superior to using a single modality. Moreover, Cole [13] adopted Lasso regression on six image modalities (T1, T2, susceptibility-weighted imaging (SWI), diffusion-MRI (dMRI), task fMRI (tfMRI), and rsfMRI) for the prediction of brain age. They have also verified that multi-modality imaging is more accurate than single-modality imaging.

In recent years, researchers have begun to investigate the impact of ML algorithms on the prediction performance of brain age. Structural imaging features have been widely used in brain age prediction studies due to their effectiveness in characterizing brain morphology. Lombardi et al. [14] compared the performance of several ML strategies, including deep neural networks (DNN), random forest (RF), SVR, and Lasso, based on the anatomical features of 2168 participants. Their results showed that DNN outperformed the other methods. Valizadeh et al. [27] applied multiple linear regression (MLR), ridge regression (RR), neural networks (NN), k-nearest neighborhood (KNN), support vector machine (SVM), and RF to various combinations of anatomical measures of 3144 participants (aged 7–96 years). They found that the NN and SVM models performed better than the other models. In another study, Baecker et al. [18] investigated the impact of input type and model choice on brain age prediction performance using regional and voxel-based anatomical measures from 10,824 participants (aged 47–73 years). Their results showed that the input type had a greater impact on performance than model choice and that SVR, RVR, and Gaussian process regression (GPR) all performed similarly. Although these studies offer a wealth of knowledge for anatomically based brain age prediction, the findings from these studies cannot be simply extended to other imaging modalities or to a multi-modality investigation. A recent study by Niu et al. [20] compared the performance of various ML models, including RR, SVR, GPR, and DNN, using imaging features from three modalities (T1, DTI, and resting-state functional MRI) in a cohort of 839 young participants. The author found that GPR, using multi-modal features, achieved the highest prediction accuracy, while the other three ML algorithms exhibited similar performance. They also suggested that multi-modality imaging features may confer an advantage for age prediction. It should be noted, however, that the study’s age span (8–21 years) limits the generalizability of the results. When examining the effects of ML algorithms, age range is a critical consideration. Brain age prediction models are typically developed for three age groups: childhood through adolescence, middle age through old age, and all ages. The predicted brain’s age can reveal how various diseases and cognitive activities have impacted the brain throughout a person’s life, and prediction errors may vary among different age groups. Infancy through adolescence is the most precise age range for prediction, followed by middle age and old age, while the entire age range is the most challenging. In most studies covering the entire age range, the majority of participants are young, with fewer middle-aged and older individuals, making it easier to predict. Therefore, it is not appropriate to directly compare the results based on different age groups or different age distributions within full-age groups. In this study, we focus on predicting the brain age of the middle-aged and senior age group, using a sizable dataset of over 10,000 individuals. To our knowledge, this is the first comprehensive exploration of the relationships between ML algorithms, imaging modalities, and brain age prediction in this age group.

To investigate the extent to which the algorithm, the imaging modality, and the interaction between them impact brain aging prediction performance, we conducted a comprehensive experiment using six ML algorithms (Lasso, RVR, SVR, XgBoost, category boost (CatBoost), and MLP) and six imaging modalities (T1, T2, SWI, diffusion-weighted imaging (DWI), tfMRI, and rsfMRI). Our study was designed to accomplish four goals. The primary goal of this study was to determine which ML algorithm could most accurately predict brain age. The second goal was to identify the imaging modality that is most sensitive to predicting brain age. Third, we sought to determine whether there was any interaction between the ML approach and image modalities. The fourth goal was to assess the interpretability of BrainAGE in multi-modal brain-age prediction.

## 2. Materials and Methods

### 2.1. Participant

UK Biobank (UKB) is a population-based prospective study of over 500,000 middle-aged and older participants (https://www.ukbiobank.ac.uk, accessed on 11 January 2021). UKB received ethical approval from the North West Multicenter Research Ethics Committee (11/NW/0382), and the study described here was approved by UKB under application number 68382. We excluded participants who had neurological or psychiatric diseases according to the International Classification of Diseases, Tenth Revision (ICD-10). For the remaining 388,721 participants, we selected individuals who had undergone brain scans using all six modalities. In total, 27,842 individuals were included in the study. The UKB participant inclusion chart is shown in Figure 1. The chosen samples were randomly divided into a training set and a test set with a ratio of 1:1. The demographic information for training and test sets was summarized in Table 1. The distributions of chronological ages of the training, test sets, as well as the entire cohort, are shown in Figure 2. Five-fold cross validation was applied to the training set for model validation.

### 2.2. Imaging Derived Phenotypes (IDPs)

The UKB Imaging Working Group (www.ukbiobank.ac.uk/expert-working-groups, accessed on 11 January 2021) and a panel of brain imaging experts designed the imaging protocols. The MRI provides multiple imaging modalities that offer complementary information. To ensure data compatibility, three imaging centers are equipped with identical scanners and fixed platforms (without significant software or hardware updates throughout the study). Each center uses a 3T Siemens Skyra with software platform VD13 and a 32-channel receive head coil for brain imaging (Skyra 3T, Siemens Healthcare Gmb H, Erlangen, Germany). Table 2 summarizes the key acquisition parameters for each modality. The order of acquisition was optimized to consider subject compliance, assuming that subject motion might increase over the scan. Therefore, the T1 was acquired first due to its central importance; for example, the processing pipeline cannot run without it. Furthermore, assuming subject wakefulness might decrease, the fMRI was also acquired early. The order of acquisition is: T1, rsfMRI, tfMRI, T2, dMRI, SWI. Additional protocol details are available at http://biobank.ctsu.ox.ac.uk/crystal/refer.cgi?id=2367, accessed on 11 January 2021, and a more in-depth description of post-processing pipelines and data outputs can be found at https://biobank.ctsu.ox.ac.uk/crystal/crystal/docs/brain_mri.pdf, accessed on 11 January 2021. To enhance the quality and scope of the image-derived phenotypes (IDPs) produced, a variety of software tools were utilized in the data release’s processing pipeline. All of these tools are freely available for use. While the FMRIB Software Library (FSL) served as the primary source of tools for the pipeline, other methods and software were also utilized. For example, one high priority is to adapt the Human Connectome Project pipelines to provide cortical surface modeling. The goal is for non-imaging experts to be able to use the IDPs directly without becoming experts in the complexities of data processing.

In the current study, IDPs [28] created by UKB were employed. They were chosen by looking through the data showcase (http://biobank.ctsu.ox.ac.uk/crystal/index.cgi, accessed on 11 January 2021). The preprocessing details of those IDPs were described in https://biobank.ctsu.ox.ac.uk/crystal/crystal/docs/brain_mri.pdf, accessed on 11 January 2021. Table 2 provides a brief summary of the definition of IDPs. The IDPs from six modalities were divided into seven feature sets, two of which (Freesurfer-based and FSL-based) were derived from T1 MRI.

### 2.3. Non-Imaging Derived Phenotypes (Non-IDPs)

During the visit, UKB participants were asked to complete a touchscreen questionnaire, participate in a verbal interview, and undergo a series of physical measurements to provide sociodemographic, lifestyle, and health-related information. Four sub-categories of data (recruitment, touchscreen, verbal interview, and physical measurements) collected were regarded as non-IDPs (Table 3), which included 814 non-IDPs. These non-IDPs were automatically curated using the FMRIB UKBiobank Normalisation, Parsing And Cleaning Kit (https://git.fmrib.ox.ac.uk/fsl/funpack, accessed on 8 November 2022) software, which automatically sorts variables into hand-curated groups and ensures that quantitative variable codings are parsed into monotonically meaningful values while separating categorical variables into multiple binary indicators. The resulting numeric vectors for all non-IDPs (quantitative and categorical) enabled easy calculation of correlation coefficients. Non-IDPs that were not related to the participant characteristics (such as seating box height, UKB ID 3077) were eliminated, and categorical non-IDPs that could not be recoded as ordered variables were also removed. Additionally, non-IDPs that were not gathered during image acquisition were excluded. Missing data were also addressed, and non-IDPs with more than 20% missing data were removed. For non-IDPs with less than 20% missing data, mean imputation was used to fill in the missing values. Finally, a total of 217 non-IDPs were retained for the upcoming statistical analysis.

### 2.4. ML Models

Chronological age and the combination of IDPs were regarded as the dependent and independent variables, respectively, for each ML prediction model. Participant age was rounded to the nearest full month. A grid search with five-fold cross validation was employed for the hyper-parameter search.

#### 2.4.1. Lasso

Lasso is a regression method proposed by Tibshirani [29] to address the problems of over-fitting and multicollinearity in ordinary least square regression (OLS). The penalty regularization parameter alpha in Lasso is responsible for regulating the penalty’s severity. The higher the value, the stronger the penalty for each parameter, which results in greater shrinkage of the coefficient sizes. The grid search space for the parameter alpha was specified as (0.001, 0.01, 0.1, 1, 10, 100).

#### 2.4.2. RVR

RVR [30] is a Bayesian framework for learning sparse regression models. In practice, RVR has better generalization capabilities and is more resilient to outliers because it employs fewer support vectors than SVR [31]. RVR uses a Bayesian framework to automatically optimize hyper-parameters, so no hyper-parameter tuning is required [5].

#### 2.4.3. SVR

The SVM [32] algorithm creates a hyperplane with the largest gap between positive and negative instances in the feature space. For data that can be linearly separated, linear SVM is frequently employed. The SVR is a regression analysis model based on the SVM. C is used to set the level of regularization. The grid search space for the parameter C was defined as (2^−7^, 2^−5^, 2^−3^, 2^−1^, 2^0^, 2^1^, 2^3^, 2^5^, 2^7^).

#### 2.4.4. XgBoost

XgBoost, which has frequently appeared in winning solutions in Kaggle competitions, is an implementation of gradient boosted decision trees [33]. To obtain the best XgBoost model, grid-searching was performed on parameter combinations of learning-rate, maximum_depth of the tree, and number of estimators. The search range was (100, 300, 500, 1000), (2, 10, 1), and (0.01, 0.03, 0.05, 0.1, 0.3, 0.5) for the number of estimators, maximum depth of the tree, and learning-rate respectively.

#### 2.4.5. CatBoost

CatBoost is also an algorithm for gradient boosting on decision trees [34]. It offers a new strategy for handling categorical features that can address the gradient bias and prediction shift issues. Grid-searching was used in learning-rate, maximum depth of the tree, and number of estimators were estimated. The search range was (100, 300, 500, 1000), (2, 10, 1), and (0.01, 0.03, 0.05, 0.1, 0.3, 0.5) for the number of estimators, maximum depth of the tree, and learning-rate, respectively.

#### 2.4.6. MLP

MLP is a feed-forward artificial neural network (ANN) that is trained using a back-propagation algorithm [35]. An MLP is composed of input nodes at each layer that form a directed graph between the output and input layers. An MLP is a neural network that connects many layers in a directed graph, which means that the data signal is routed through the nodes of the graph in only one direction. The MLP model in our study contains four fully connected layers, and the number of neurons in the hidden layers is 1024, 512, 256, and 128 respectively. The search range was (0.00001, 0.00002, 0.00004, 0.00008, 0.00016, 0.00032, 0.00064, 0.00128, 0.00256, 0.00512) for learning-rate.

### 2.5. BrainAGE

The difference between Predicted age and chronological age is the BrainAGE score (Equation (1)).
(1)BrainAGE=Predicted age−chronological age

### 2.6. Bias Correction in Brain Age Prediction

Predicted brain ages obtained from regression models are subject to the phenomenon of “regression toward the mean” [36], which is an inherent statistical phenomenon that leads to a bias in predicted brain age. This phenomenon can lead to an overestimation of brain age for younger individuals and an underestimation of brain age for older individuals in relation to chronological age. Several previous studies [13,37,38] have reported a negative correlation between the difference between predicted brain age and chronological age. To address this issue, corrected brain age was calculated using Equation (2) [39].
(2)$${\mathrm{Predicted}\;\mathrm{age}}_{\mathrm{corrected}}={\mathrm{Predicted}\;\mathrm{age}}_{\mathrm{raw}}-\beta -\alpha *\mathrm{Chronological}\;\mathrm{age}$$
where ${\mathrm{Predicted}\;\mathrm{age}}_{\mathrm{raw}}$ is the predicted brain age, and *α* and *β* are the slope and intercept of the regression line indicting the relationship between BrainAGE and chronological age obtained from the training set. Applying Equation (2) to the test set results in the corrected predicted brain age, which is ${\mathrm{Predicted}\;\mathrm{age}}_{\mathrm{corrected}}$.

### 2.7. Ensemble Learning

This study employed ensemble learning, a ML technique that combines multiple models to improve prediction performance. Six multi-modality image-based bias-corrected brain ages were averaged to create the final predicted brain age, which utilizes the strengths of each individual model to reduce the effects of bias and variance in the predictions.

### 2.8. Statistical Analysis

Statistical analysis was performed using the SPSS 26 software (SPSS, 1989; Apache Software Foundation, Chicago, IL, USA). One of the primary analyses we conducted was a two-way ANOVA, which allowed us to examine the impact of the feature set, the ML algorithm, and the combination of the feature set and the ML algorithm on the classification performance of uni-modality brain age prediction models. Additionally, we used one-way ANOVA to compare several ML algorithms for the multi-modality brain age prediction model. The statistical significance level was set at *p* < 0.001.

To further understand the potential relationships between the non-IDPs and the bias-adjusted BrainAGE, we computed Pearson correlations between the bias-adjusted BrainAGE and 217 non-IDPs. To address the issue of multiple comparisons, we employed a false discovery rate (FDR) of *q* < 0.01. Only non-IDPs with a certain correlation (*r*^2^ ≥ 0.25%) with BrainAGE were retained based on the correlation coefficient.

The interpretability of brain age models relies on retaining more interpretable components in the BrainAGE variance and filtering out noise. However, different machine learning models obtain BrainAGE with different variances, and the BrainAGE from a model that is highly correlated with more non-IDPs does not guarantee better interpretability of the model. This may be due to the model retaining more noise while preserving more interpretable components. If the signal-to-noise ratio in BrainAGE is low, this may lead to some false positive results. Therefore, we included some non-IDPs that have been repeatedly identified in BrainAGE studies, such as diastolic blood pressure [13,40,41], systolic blood pressure [42,43,44], alcohol intake [45,46,47], a diabetes diagnosis [48,49,50], and smoking status [51,52,53], in the comparison to observe the model’s interpretability of these typical findings.

## 3. Results

### 3.1. Brain Age Prediction Models

We applied six ML models to uni-modality and multi-modality brain imaging features. Table 4 provides a summary of the performance of the predictions. The Freesurfer-based features from T1 have the highest performance, followed by the DWI features and the FSL-based features from T1. SWI’s prediction performance was the worst among all the features. When ML models from six modalities were compared, MLP produced the greatest results in FSL, T2, rsfMRI, and tfMRI. Lasso had the best performance in Freesurfer. The best DWI performance was obtained by SVR. In SWI, the MAE was 6.253 years for Lasso and SVR. We found that the performance of multi-modality models was overall superior to that of any model trained with uni-modal data. Incorporating IDPs from six modalities, Lasso had the highest prediction accuracy (MAE = 2.741 years). Using multi-modal brain imaging features, the prediction accuracy of RVR (MAE = 2.767 years), SVR (MAE = 2.860 years), CatBoost (MAE = 2.970 years), and MLP (MAE = 2.857 years) was close to that of Lasso.

Figure 3 shows the correlation matrix for chronological age and estimated brain age of six imaging modalities via six ML algorithms. The best imaging feature sets (i.e., those with the highest correlation with chronological age and the lowest MAE) are the Freesurfer-based features from T1, followed by DWI features and FSL-based features from T1. Estimated brain age based on features from T2, rsfMRI, and tfMRI only had a weak or moderately positive association with chronological age. The models based on SWI features had the worst performance. In Lasso, the predicted age is equivalent to a constant value, and the variance of the estimated age is zero, so the correlation coefficient of chronological age and estimated brain age becomes infinity. This is labeled as “?” in the correlation matrix. Because models based on SWI, T2, rsfMRI, or tfMRI are weak predictors of brain age, the two-way ANOVA analysis only used three feature sets from T1 and DWI. Our study employed a two-way ANOVA, which revealed a significant main effect of image modality (F (2,10) = 1289.699, *p* < 0.001), a main effect of ML algorithms (F (5,10) = 45.997, *p* < 0.001), and a significant interaction effect (F (10,10) = 11.429, *p* < 0.001). However, the effects of the ML algorithm and the interaction effect were small, accounting for only 8.9% and 4.4% of the variance explained by the feature set, respectively. Subsequent post hoc tests showed that the ML models trained with the Freesurfer-based feature set outperformed those trained with the FSL-based feature set (*p* < 0.001) or the DWI-based feature set (*p* < 0.001). Similarly, the ML models trained with the DWI-based feature set also outperformed those trained with the FSL-based feature set (*p* < 0.001).

### 3.2. Leave-One-Modality-Out Analysis

To further explore the effects of different imaging modalities on brain prediction, we removed one of the feature sets in multi-modal modeling. Here, each feature set was removed once, and the six ML models were used to train and test the remaining feature sets (Table 5). Removing SWI-, T2-, rsfMRI-, or tfMRI-based features almost had no effect on model performance. The largest performance degradation was observed when the Freesurfer-based features were excluded. It is worth noting that there is a slight performance improvement for all models except Lasso, RVR, and CatBoost when SWI-based features are excluded.

### 3.3. Brain Age Bias Correction for Multi-Modality Models

This gave the result of brain age bias correction (see Appendix A). After bias correction, the red line became closer to the gray line (the identity line), which indicated the correction of the age-related bias.

Figure 4 shows the correlation matrix of chronological age and estimated brain age of multi-modal images via six ML algorithms. The brain ages predicted by these ML models are highly correlated not only with each other but also with chronological age.

Table 6 shows the slope and R^2^ of the predicted brain age versus the actual age before and after bias correction. Lasso and SVR had a slope of 0.98, RVR had a slope of 0.99. The rest of the models had a slope of around 0.90. R^2^ estimates the proportion of the predicted age variance that can be explained by chronological age. The explanatory power of R^2^ for Lasso is 85%, followed by SVR, MLP, CatBoost, XgBoost, and RVR. The absolute errors with different ML approaches were significantly different based on a one-way ANOVA (F (5, 83,046) = 46.443, *p* < 0.001) followed by the Bonferroni post hoc test. The absolute errors of the Lasso approach were significantly lower than all other ML models (*p* < 0.001). We also averaged the corrected estimated brain ages of the six ML models, and the MAE was 2.338 years.

### 3.4. BrainAGE Variance Explained by Non-IDPs

To facilitate interpretation of the BrainAGE, the relationships between the BrainAGE and the non-IDPs are illustrated in Figure 5. There is a total of 128 non-IDPs in the figure, and at least one of the seven models for those non-IDPs meets the FDR *q* < 0.01 and *r*^2^ ≥ 0.25% requirement. The positive correlation between a non-IDP and BrainAGE implies that accelerated brain aging is associated with higher levels of this non-IDP, suggesting that the higher value of this non-IDPs is “bad” for brain maintenance.

To further evaluate the interpretability of the BrainAGE, we investigated its sensitivity to five non-IDPs commonly identified in BrainAGE research. We ranked the performance of seven ML algorithms based on their correlation coefficient (*r*) with these variables, as shown in Figure 6. We then calculated a comprehensive ranking to provide an overall assessment of each model’s performance. Notably, we observed varying degrees of sensitivity to these non-IDPs, among the ML models, suggesting that their impact on the BrainAGE is not uniform. Our results indicate that Lasso regression and ensemble learning models achieved the highest rank in the comprehensive evaluation.

## 4. Discussion

In this study, we aimed to investigate the impact of the ML algorithm, image modalities, and the interaction between the two on the performance of brain age prediction. To achieve this, we employed imaging data from six modalities to create 2218 IDPs. We evaluated a total of seven ML models, including six individual models and one ensemble model. Our study had several objectives: first, to identify the ML method that could estimate brain age most accurately; second, to determine which imaging modality was most effective in predicting brain age; third, to examine how different ML algorithms interact image features on brain aging prediction; and finally, to assess the interpretability of BrainAGEs generated by various ML models.

### 4.1. Image Modalities and ML Approaches

While all imaging modalities do demonstrate some ability to predict brain aging, they are not equally effective. T1 and DWI were determined to be the most relevant image modalities for brain age prediction. We found that changes in gray matter morphology and WM microstructures, particularly cortical thickness measurements, are the most critical imaging features. This conclusion was also supported by the leave-one-modality-out experiment. There are two reasons for the superior performance of T1 and DWI in predicting brain age. First, cognitive impairment in older adults is often related to brain atrophy and myelin degradation, as demonstrated by clinical and neuropathological studies [54,55]. T1-weighted images offer high anatomical resolution, allowing for detailed visualization of brain structures. Meanwhile, DWI images are sensitive to microstructural changes in white matter, providing information on white matter connectivity and integrity, which decline with age. This makes T1 and DWI-based IDPs particularly relevant for predicting brain age. Second, compared to the other modalities, T1 (N = 1436) and DWI (N = 675) had considerably more IDPs. Increasing the number of features in an imaging modality has been shown to improve the predictive ability of ML models. This is because a greater number of features can capture more detailed information about the biological and pathological processes underlying the imaging data, which can improve the accuracy of the model’s predictions. However, it is also important to consider the potential for overfitting when using large feature sets, as this can lead to reduced generalizability and poorer performance on new data. The other four modalities were only able to explain a small amount of variance in age, particularly the SWI and tfMRI. Among seven feature sets, the FSL-based and Freesurfer-based feature sets had the highest correlation with predicted age (r ranging from 0.74 to 0.84 on six ML models). Our findings are consistent with the findings of previous studies, which have shown gray matter morphology [4,15,19] or T1 integrity [25] to be reliable predictors of brain age. Surface-based features showed promise, with all ML models achieving MAEs below 3.48 years, and the lowest MAE being 3.127 years. Surface-based features have several advantages over volumetric measures in assessing age-related changes in the brain. Surface-based features are more accurate and precise than volumetric measures in capturing age-related changes in the brain [56,57]. Additionally, surface-based features are better at detecting local changes in brain morphology and handling partial volume averaging [58]. Furthermore, surface-based features have a higher sensitivity to age-related changes and are a reliable predictor of chronological age [59]. The performance rankings of several ML methods for a given set of image features are relatively similar, and the imaging modality is more important than the choice of ML models. The results of this study were based on cognitively normal participants. However, under disease conditions, the imaging modality might play a more critical role. For example, in patients with WM disease, a DWI-based brain age is more meaningful than a cortical thickness-based brain age.

Even though features from T1 or DWI have shown promising results, adding additional modalities may lead to a more efficient prediction than any single modality [13], despite their noticeable collinearity. When different ML algorithms were applied to construct predictors using the features from all imaging modalities, the multi-modal models outperformed those trained with uni-modal models. One potential confounding factor in age prediction models is the shared variance between chronological age and BrainAGE. To address this issue, an age bias adjustment approach was used to remove the shared variance between chronological age and BrainAGE. After correction, there was a strong positive correlation between the predicted ages from different models (*r* ranging from 0.93 to 1). High positive correlations between chronological age and predicted age (*r* ranged from 0.88 to 0.92) were also observed. These models contributed almost 75% of the variance in the test data, with a corrected MAE of less than 2.71 years and a minimum of 2.45 years, and a corrected R^2^ greater than 79.8% and a maximum of 85.0%.

In this study, the Lasso model outperformed other ML models in terms of prediction accuracy, likely due to its ability to handle high-dimensional multi-modal IDPs, prevent overfitting, and effectively select important features while removing irrelevant or redundant ones, while also addressing collinearity among predictor variables. Our results show that the performance of the Lasso model is comparable to or even better than that reported in earlier studies (with MAEs ranging from 3.4 years to 5.99 years) [13,14,15]. Unlike previous studies, we utilized six image modalities and a larger sample size (N = 27,842) in our investigation. Ensemble learning is an ML technique that combines predictions from multiple models to produce a more accurate prediction. In our study, the ensemble model (MAE = 2.338 years) outperformed the single ML model, indicating the benefits of ensemble learning. The improved performance of the ensemble model is attributed to its ability to leverage the strengths of multiple models and mitigate their weaknesses, leading to overall better performance. Additionally, ensemble learning enhances the stability of predictions by reducing the variance of individual models and lowering the risk of overfitting, a common issue in ML models.

### 4.2. The Interpretability of BrainAGE

Brain age prediction is an important area of research, as it provides a biomarker for cognitive aging and age-related neurological diseases. To translate this research tool into a clinical application, it is crucial to identify the factors that underlie the BrainAGE biomarker in an interpretable manner. Brain age prediction models strive to accurately estimate the age of the brain while also providing a high degree of interpretability in the BrainAGE. The MAE is a commonly used metric to evaluate the accuracy of brain age predictions. Interestingly, as the predictive error decreases, there is an initial increase and then decrease in the interpretability of the BrainAGE. If predictive error is zero, the BrainAGE is also zero, indicating a perfect match between predicted and chronological age, but this also means there is no interpretability. Thus, finding a balance between accuracy and interpretability is crucial for developing effective brain age prediction models with practical applications in clinical and research settings.

One possible approach to improve the interpretability of BrainAGE is to enhance the signal-to-noise ratio by reducing unexplained variance while preserving explained variance. However, this approach cannot be implemented in the error function as interpretability is a population concept. To address this issue, we investigated the interpretability of BrainAGE from two perspectives. Firstly, we conducted a correlation analysis between BrainAGE predicted by seven ML models and 217 non-IDPs. Interestingly, we found that the BrainAGE generated by CatBoost and XgBoost had closer associations with non-IDPs, but they also had greater prediction errors in brain age prediction. However, a closer relationship with a large number of non-IDPs does not necessarily indicate stronger interpretability but may be due to a larger variance in BrainAGE, which can lead to more false positive reports. Secondly, we investigated the association between BrainAGE predicted by seven ML models and five non-IDPs previously identified in BrainAGE studies. Our results showed that Lasso regression and ensemble learning models ranked highest in the comprehensive evaluation of the relationship between BrainAGE and five non-IDPs. As there is a non-linear relationship between model error and interpretability, reducing model error may not always improve interpretability. Nevertheless, our study shows the critical role of minimizing model errors in enhancing the interpretability of BrainAGE. Of the six ML models evaluated, Lasso demonstrated the lowest model prediction error and ranked highest in the comprehensive evaluation with the five BrainAGE-related factors. Therefore, Lasso should be the preferred choice. A single ML model often has limitations and may not perform well on all datasets. This was evident in the study where no single model showed the strongest correlation with all five factors. In contrast, ensemble learning combines the predictions of multiple models to achieve a more robust and accurate prediction. By using diverse models, ensemble learning can capture different aspects of the data, resulting in a higher explainable variance. In the study, ensemble learning reflected the combined ability of the models to explain the five factors, resulting in a lower model prediction error than Lasso and comparable performance in the comprehensive evaluation. Moreover, the ensemble learning model exhibited stable performance in terms of correlations for non-IDPs, ranking in the middle of the pack of seven models. Therefore, we recommend the use of ensemble learning when computational efficiency is not a major concern, given its superior performance in the comprehensive evaluation and lower model prediction error compared to Lasso, as well as its relatively stable performance with respect to non-IDPs.

### 4.3. Limitation

The present study has several significant limitations. Firstly, the study population included both healthy individuals and those with diseases or risk factors, which may have altered brain structure or function, and decreased sensitivity to aging. Secondly, this study examines brain age prediction using data from the UKB database, specifically focusing on individuals of white British ancestry. However, the generalizability of these findings is limited by the study’s exclusive focus on this specific population. Further research is necessary to understand how brain age prediction may vary across different ethnic and national groups. Thirdly, it is important to note that all the study’s findings were based on globally and locally customized imaging features from several free neuroimaging tools. The deep learning-based framework can identify the optimal representation of features from a high-dimensional image space, eliminating the need for domain knowledge in feature engineering.

## 5. Conclusions

The present study conducted a systematic and rigorous evaluation of six ML algorithms applied to the UKB dataset for the purpose of brain age prediction. Our findings demonstrate that all six models examined are suitable for this task. Our analysis also revealed that T1-weighted MRI and DWI are generally the most informative image modalities for brain age prediction. Furthermore, our findings suggest that image modality is more important than ML algorithm selection in determining the accuracy of brain age prediction. While we observed interaction effects between the image feature set and the ML algorithm, these effects were found to account for only a small variance and should be interpreted with caution. As expected, the multi-modality model outperformed the unimodality models. Among the six ML algorithms, the Lasso model was found to deliver the best outcomes for MAE in multi-modality brain age prediction and ranked highest in comprehensive evaluation with five widely proven BrainAGE-related factors. However, our analysis also revealed that the ensemble learning model outperformed Lasso and should be the choice when computational efficiency is not a critical factor. Overall, our study offers valuable insights into the effectiveness of different ML algorithms in predicting brain age among middle-aged and older adults. Our findings have significant implications for future research endeavors aimed at enhancing the accuracy of brain age prediction and deepening our comprehension of age-related changes in the brain.

## Figures and Tables

**Figure 1 sensors-23-03622-f001:**
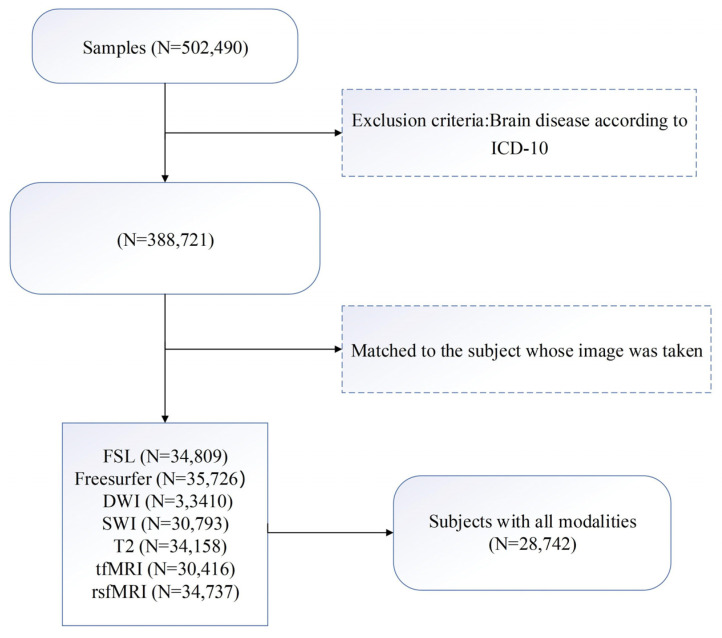
UKB participant inclusion chart.

**Figure 2 sensors-23-03622-f002:**
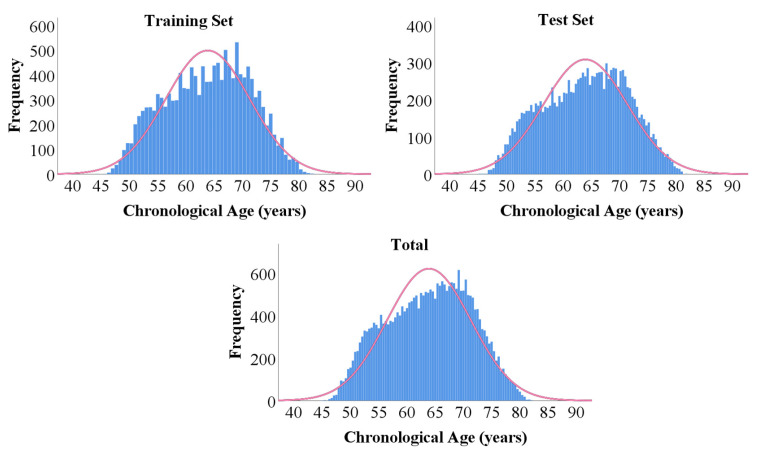
The distributions of chronological ages of the training, test sets, and whole cohort.

**Figure 3 sensors-23-03622-f003:**
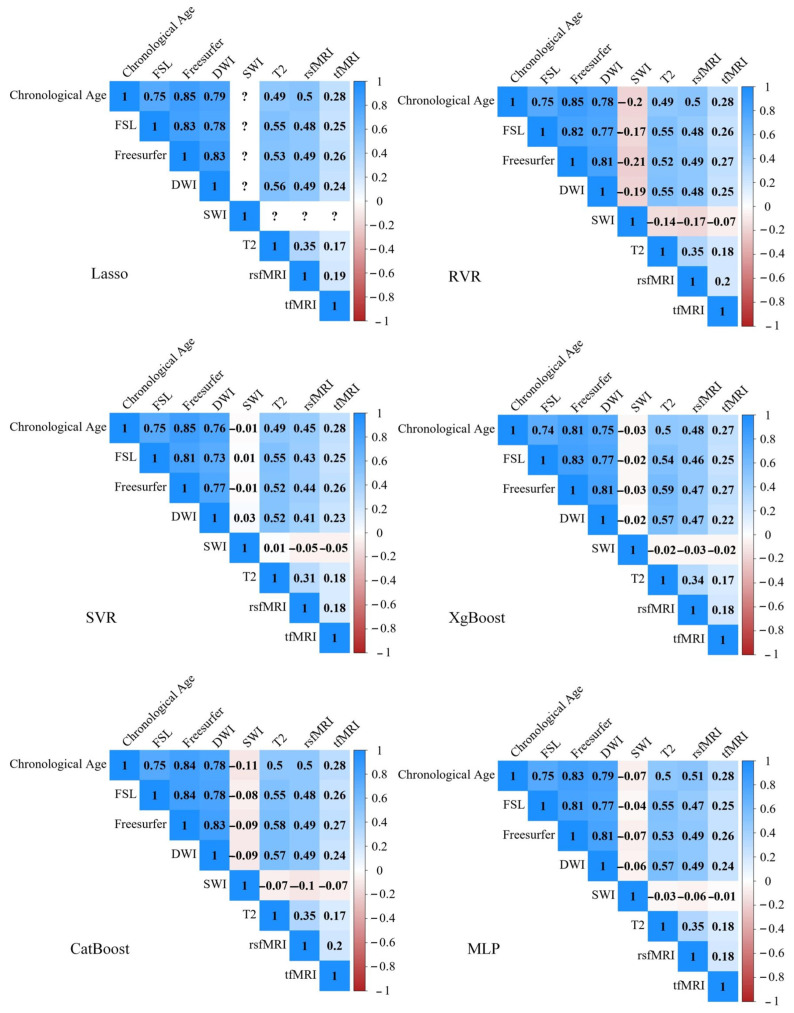
Pearson correlation matrix of chronological age and estimated brain age of six imaging modalities via six ML algorithms.

**Figure 4 sensors-23-03622-f004:**
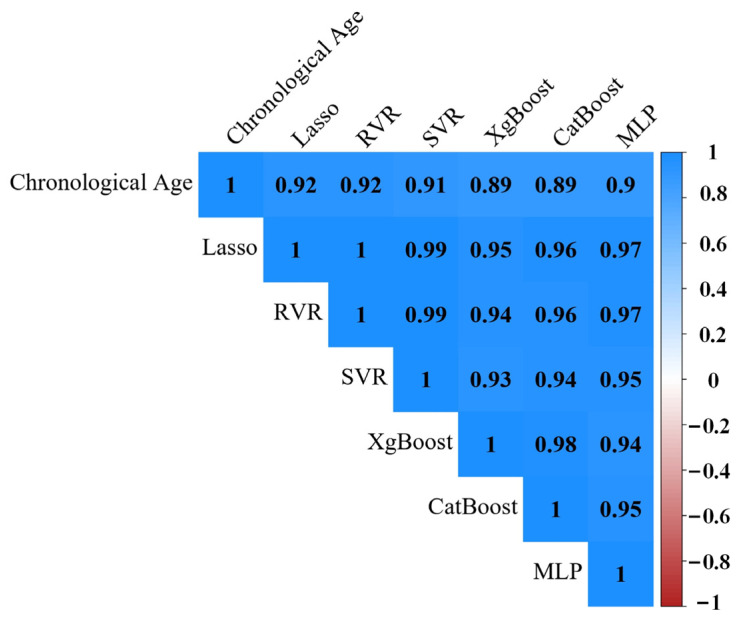
Pearson correlation matrix of chronological age and estimated brain age of multimodality via six ML algorithms.

**Figure 5 sensors-23-03622-f005:**
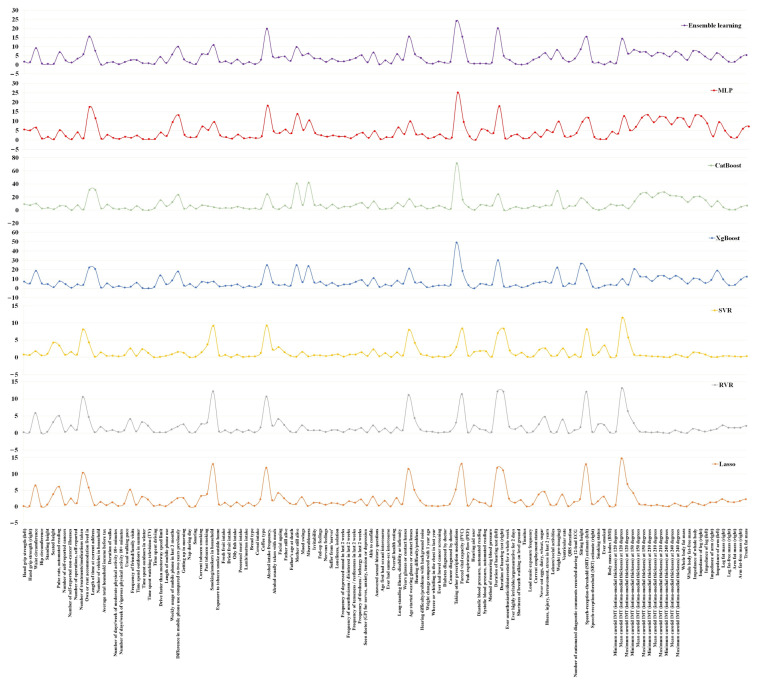
A Manhattan plot of -log 10 of FDR corrected p values (*y* axis) by non-IDPs (*x* axis).

**Figure 6 sensors-23-03622-f006:**
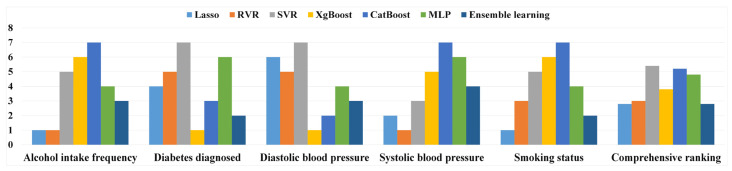
Ranking results of seven ML model on five non-IDPs.

**Table 1 sensors-23-03622-t001:** Demographic information for participants included in the training and the test sets.

Demographic Information	Training Set	Test Set	Total
Number of participants	14,000	13,842	27,842
Age (mean (SD))	63.8 (7.5)	63.9 (7.5)	63.9 (7.5)
Sex (Male/Female)	6629/7371	6494/7348	13,123/14,719

**Table 2 sensors-23-03622-t002:** Description of IDPs.

Modality		Imaging Protocol	Description of IDPs	UKB ID	IDPs	N
T1	FSL	Three-dimensional scrambled phase gradient echo sequence, image matrix 208 × 256 × 256 mm^3^, TI/TR = 880/2000 ms, voxel resolution 1 × 1 × 1 mm^3^	anatomical measures of brain structures	25000~25024, 25782~25920	164	40,680
	Freesurfer			26501~27772	1272	43,075
DWI		Planar echo imaging, image matrix 104 × 104 × 72 mm^3^, TE/TR = 92/3600 ms, voxel resolution 2 × 2 × 2 mm^3^. Two *b* values (*b* = 1000, 2000 s/mm^2^), 100 different directions in total, multi-band acceleration factor of 3	the integrity of micro-structural tissue compartments and structural connectivity between pairs of brain regions	25056~25730	675	39,022
SWI		3D dual-echo gradient echo sequence, TE1/TE2/TR = 9.4/20/27 ms, image matrix 256 × 288 × 48 mm^3^, voxel resolution 0.8 × 0.8 × 3 mm^3^	venous vasculature, microbleeds or aspects of micro-structure	25026~25039	14	35,937
T2		Liquid decay inversion recovery sequence, image matrix 192 × 256 × 256 mm^3^, TI/TR = 1800/5000 ms, voxel resolution 1.05 × 1 × 1 mm^3^	the volume of WM lesions	25781	1	39,898
rsfMRI		Planar echo imaging, image matrix 88 × 88 × 64 mm^3^, TE/TR = 39/735 ms, voxel resolution 2.4 × 2.4 × 2.4 mm^3^, total 490 time points	the apparent connectivity between pairs of brain regions, and the amplitude of spontaneous fluctuation within each region	25754~25755	76	40,594
tfMRI		Planar echo imaging, image matrix 88 × 88 × 64 mm^3^, TE/TR = 39/735 ms, voxel resolution 2.4 × 2.4 × 2.4 mm^3^, total 332 time points	the strength of response to the specific task within a given brain mask	25040, 25042, 25044, 25046, 25048, 25050, 25052, 25054, 25761~25768	16	35,499
total		/	/	/	2218	27,842

**Table 3 sensors-23-03622-t003:** Description of non-IDPs.

Non-IDPs	Description
Recruitment	Contains information about a participant’s arrival at the assessment center and the locations from which they were recruited.
Touchscreen	Contains information from the touchscreen questionnaire completed at the assessment center and is divided into several sub-categories (sociodemographics, lifestyle and environment, early life factors, family history, psychosocial factors, health and medical history, and sex-specific factors).
Verbal interview	Contains information based on a verbal interview conducted by trained staff at the assessment center and is divided into several sub-categories (early life factors, employment, medical conditions, medications, and operations).
Physical measures	Contains information from physical measurements performed at the assessment center and is divided into sub-categories based on the type of physical measurement performed (blood pressure, carotid ultrasound, hearing test, hand grip strength, anthropometry, bone-densitometry of heel, spirometry, ECG at rest, and 12-lead ECG).

**Table 4 sensors-23-03622-t004:** Prediction performance of six ML models.

	MAE (Years)							
Method	T1		DWI	SWI	T2	rsfMRI	tfMRI	All Modality
	FSL	Freesurfer						
Lasso	3.945	3.127	3.587	6.253	5.301	5.281	5.969	2.741
RVR	3.947	3.149	3.682	6.267	5.301	5.279	5.953	2.767
SVR	3.957	3.180	3.576	6.253	5.288	5.286	5.955	2.860
XgBoost	3.999	3.480	3.955	6.261	5.276	5.368	5.975	3.222
CatBoost	3.930	3.265	3.742	6.256	5.272	5.300	5.956	2.970
MLP	3.883	3.287	3.600	6.264	5.267	5.248	5.941	2.857

**Table 5 sensors-23-03622-t005:** Brain age prediction performance, leaving out single modality.

	MAE (Years): Excluded Modality								
Method	T1		T1	DWI	SWI	T2	rsfMRI	tfMRI	All Modality
	FSL	Freesurfer							
Lasso	2.817	3.132	3.427	2.910	2.752	2.741	2.786	2.753	2.741
RVR	2.838	3.154	3.423	2.938	2.780	2.769	2.826	2.786	2.767
SVR	2.918	3.138	3.415	2.988	2.851	2.862	2.891	2.873	2.860
XgBoost	3.311	3.477	3.758	3.301	3.218	3.238	3.271	3.222	3.222
CatBoost	3.049	3.285	3.558	3.096	2.971	2.975	3.004	2.970	2.970
MLP	2.945	3.165	3.447	3.044	2.807	2.876	2.924	2.857	2.857

**Table 6 sensors-23-03622-t006:** Corrected slope vs. R^2^.

Model	Corrected Slope	Uncorrected R^2^	Corrected R^2^	Delta R^2^	Uncorrected MAE	Corrected MAE
Lasso	0.980	0.787	0.850	0.063	2.741	2.450
RVR	0.990	0.784	0.847	0.063	2.767	2.476
SVR	0.980	0.770	0.830	0.060	2.860	2.577
XgBoost	0.890	0.704	0.798	0.094	3.222	2.708
CatBoost	0.860	0.747	0.800	0.053	2.970	2.673
MLP	0.910	0.767	0.810	0.043	2.857	2.647

## Data Availability

The imaging datasets generated by UK Biobank analyzed during the current study are available via the UK Biobank data access process (see http://www.ukbiobank.ac.uk/register-apply/, accessed on 11 January 2021). The UK Biobank’s Research Access Administration Team handles all data access requests from academic and commercial researchers without any preference or exclusivity. The requests are evaluated based on whether they support health research in the public interest, and if so, they are approved rapidly. Detailed information about the data available from UK Biobank is available at http://www.ukbiobank.ac.uk, accessed on 11 January 2021. The exact number of participants with imaging data currently available in UK Biobank may differ slightly from those described in this paper.

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
