# Peer review of "Comparison of Machine Learning Models for Brain Age Prediction Using Six Imaging Modalities on Middle-Aged and Older Adults"

_sensors, 2023, doi:10.3390/s23073622_

Round 1

Reviewer 1 Report

Machine learning (ML) is extremely important for medical imaging and is an essential part of clinical decision support systems. It is especially relevant for neuroimaging where image interpretation and analysis is a complicated task for radiologists. The manuscript presents valuable insights into the development of accurate and reliable brain age prediction models based on multiple imaging modalities.

In my opinion the paper generally well written and well structured, the list of references is adequate to the current state of this field of research. 

In my opinion research topics and content of the manuscript are considered a little more appropriate to be published in biomedical journals, e.g., Biomedicines, but are undoubtedly deserved to be published in Sensors.

Authors present a large amount of research results in form of bulky tables and figures (for example, figure 5,6), which are meaningful and relevant, but often hard to read. I can suggest moving at least part of these data to supplements.

Author Response

Thank you for your valuable feedback on our manuscript. We appreciate your positive remarks about the importance of machine learning in medical imaging and the relevance of our study in predicting brain age using multiple imaging modalities.

We also appreciate your suggestion regarding the presentation of our results. We have revised our manuscript and moved some of the figures and tables to the supplemental materials to improve the readability and flow of the paper.

Regarding the choice of journal, we understand your point that our research topic may be more appropriate for biomedical journals. However, we believe that Sensors is an excellent platform to disseminate our findings as it is a leading journal in the field of sensor technology and its applications in various domains, including medical imaging.

Once again, we appreciate your feedback and suggestions, which have helped us to improve the quality of our manuscript.

Reviewer 2 Report

Overall, this is a valuable study.

Some minor comments:

1. The title needs to be modified as:

Comparison of machine learning models for brain age prediction using six IMAGING modalities on middle-aged and older adults

2. It would be helpful to insert the R^2 values (from Table 6) on the regression plots in Figure 5. 

3. The first raw of Table 2 looks confusing in terms of formatting. 

Author Response

Thank you for your valuable feedback on our manuscript. We appreciate your positive remarks about the value of our study in comparing machine learning models for brain age prediction using multiple imaging modalities on middle-aged and older adults.

Regarding your minor comments, we have addressed them as follows:

  1. We appreciate your suggestion to modify the title of our paper,  we have changed “image modality” to "imaging modalities".

  2. Thank you for your suggestion to insert R^2 values on the regression plots in Figure 5. We have updated the figure to include the R^2 values for each plot to enhance the clarity of the results.

  3. We appreciate your feedback on the formatting of the first row of Table 2. We have revised the formatting of the table to improve its clarity and readability.

Once again, we appreciate your feedback and suggestions, which have helped us to improve the quality of our manuscript.